# Estimating the false discovery risk of (randomized) clinical trials in medical journals based on published *p*-values

Ulrich Schimmack[1]*, František Bartoš[2,3]

**1** Department of Psychology, University of Toronto Mississauga, Mississauga, Canada, **2** Department of Psychological Methods, University of Amsterdam, Amsterdam, The Netherlands, **3** Institute of Computer Science, Czech Academy of Sciences, Prague, Czech Republic

* ulrich.schimmack@utoronto.ca

**Data Availability Statement:** Supplementary Materials including data and R scripts for reproducing the simulations, data scraping, and analyses are available from https://osf.io/y3gae/.

## Abstract

The influential claim that most published results are false raised concerns about the trustworthiness and integrity of science. Since then, there have been numerous attempts to examine the rate of false-positive results that have failed to settle this question empirically. Here we propose a new way to estimate the false positive risk and apply the method to the results of (randomized) clinical trials in top medical journals. Contrary to claims that most published results are false, we find that the traditional significance criterion of $\alpha = .05$ produces a false positive risk of 13%. Adjusting $\alpha$ to .01 lowers the false positive risk to less than 5%. However, our method does provide clear evidence of publication bias that leads to inflated effect size estimates. These results provide a solid empirical foundation for evaluations of the trustworthiness of medical research.

## Introduction

Many sciences are facing a crisis of confidence in published results [1]. Meta-scientific studies have revealed low replication rates, estimates of low statistical power, and even reports of scientific misconduct [2].

Based on assumptions about the percentage of true hypotheses and statistical power to test them, Ioannidis [3] arrived at the conclusion that most published results are false. It has proven difficult to test this prediction. First, large scale replication attempts [4–6] are inherently expensive and focus only on a limited set of pre-selected findings [7]. Second, studies of meta-analyses have revealed that power is low, but rarely lead to the conclusion that the null-hypothesis is true [8–16] (but see [17, 18]).

So far, the most promising attempt to estimate the false discovery rate has been Jager and Leek's [19] investigation of *p*-values in medical journals. They extracted 5,322 *p*-values from abstracts of medical journals and found that only 14% of the statistically significant results may be false-positives. This is a sizeable percentage, but it is inconsistent with the claim that most published results are false. Although Jager and Leek's article was based on actual data, the article had a relatively minor impact on discussions about false-positive risks, possibly due to several limitations of their study [20–23].

The zcurve R package is available from https://cran.r-project.org/package=zcurve.

**Funding:** The authors received no specific funding for this work.

**Competing interests:** The authors have declared that no competing interests exist.

One problem of their estimation method is the problem to distinguish between true null-hypotheses (i.e., the effect size is exactly zero) and studies with very low power in which the effect size may be very small, but not zero. To avoid this problem, we do not estimate the actual percentage of false positives, but rather the maximum percentage that is consistent with the data. We call this estimate the false discovery risk (FDR). To estimate the FDR, we take advantage of Sorić's [24] insight that the false discovery risk is maximized when power to detect true effects is 100%. In this scenario, the false discovery rate is a simple function of the discovery rate (i.e., the percentage of significant results). Thus, the main challenge for empirical studies of FDR is to estimate the discovery rate when selection bias is present and inflates the observed discovery rate. To address the problem of selection bias, we developed a selection model that can provide an estimate of the discovery rate before selection for significance. The method section provides a detailed account of our method and compares it to Jager and Leek's [19] approach.

## Methods

To estimate the false discovery rate, Jager and Leek [19] developed a model with two populations of studies. One population includes studies in which the null-hypothesis is true ($\mathcal{H}_0$). The other population includes studies in which the null-hypothesis is false; that is, the alternative hypothesis is true ($\mathcal{H}_1$). The model assumes that the observed distribution of significant $p$-values is a mixture of these two populations, modelled as a mixture of truncated beta distributions of the $p$-values. The first problem for this model is that it can be difficult to distinguish between studies in which $\mathcal{H}_0$ is true and studies in which $\mathcal{H}_1$ is true, but they had low statistical power. The second problem is that published studies are heterogeneous in power which might not be properly captured when modelling distribution of $p$-values under $\mathcal{H}_1$ with a truncated beta distribution. To avoid the first problem, we refrain from estimating the actual false discovery rate, which requires a sharp distinction between true and false null-hypotheses (i.e., a distinction between zero and non-zero effect sizes). Instead, we aim to estimate the maximum false discovery rate that is consistent with the data. We refer to this quantity as the false discovery risk. To avoid the second problem, we use z-curve which models the distribution of observed $z$-statistics as a mixture of truncated folded normal distributions. This mixture model allows for heterogeneity in power of studies when $\mathcal{H}_1$ is true [25, 26].

### False discovery risk

To estimate the false discovery risk, we take advantage of Sorić's [24] insight that the maximum false discovery rate is limited by statistical power to detect true effects. When power is 100%, all non-significant results are produced by testing false hypotheses ($\mathcal{H}_0$). As this scenario maximizes the number of non-significant $\mathcal{H}_0$, it also maximizes the number of significant $\mathcal{H}_0$ tests and the false discovery rate. For example, if 100 studies produce 30 significant results, the discovery rate is 30%. And when the discovery rate is 30%, the maximum false discovery risk with $\alpha = 0.05$ is $\approx 0.12$. In general, the false discovery risk is a simple transformation of the discovery rate, such as

$$\text{false discovery risk} = \frac{1 - \text{discovery rate}}{\text{discovery rate}} \times \frac{\alpha}{1 - \alpha}. \tag{1}$$

If all conducted hypothesis tests were reported, the false discovery risk could be determined simply by computing the percentage of significant results. However, it is well-known that journals are more likely to publish statistically significant results than non-significant results. This selection bias renders the observed discovery rate in journals uninformative [25, 26]. Thus, a

major challenge for any empirical estimates of the false discovery risk is to take selection bias into account.

## Z-curve 2.0

Brunner and Schimmack [25] developed a mixture model to estimate the average power of studies after selection for significance. Bartoš and Schimmack [26] recently published an extension of this method that can estimate the expected discovery rate before selection for significance on the basis of the weights derived from fitting the model to only statistically significant $p$-values. The selection process assumes that the probability of a study to be published is proportional to its power [25]. Extensive simulation studies have demonstrated that z-curve produces good large-sample estimates of the expected discovery rate [26]. Moreover, these simulation studies showed that z-curve produces robust confidence intervals with good coverage. As the false discovery risk is a simple transformation of the expected discovery rate, these confidence intervals also provide confidence intervals for estimates of the false discovery risk.

Z-curve obtains the expected discovery rate estimate from estimating the mean power of studies before selection of significance. Z-curve leverages distributional properties of $z$-statistics—studies homogeneous in power produce normally distributed $z$-statistics centered at a $z$-transformation of the true power (e.g., Equations 3 in Bartoš and Schimmack [26]). Z-curve incorporates the selection on statistical significance via a truncation at the corresponding $\alpha$ level, similarly to the Jager and Leek's approach. However, in contrast to other approaches, z-curve acknowledges that studies are naturally heterogeneous in power (i.e., studies are investigating different effects with different sample sizes) and estimates the mean power via a mixture model,

$$f(z; \theta) = \sum_{j=1}^{J} \pi_j f_{j[a,b]}(z; \theta_j),$$

where $z$ correspond to statistically significant $z$-statistics modelled via $J$ mixtures of truncated folded normal distributions, $f(z; \theta)$, where $\theta$ denotes parameters of the truncated folded normal distributions. See Fig 1 in Bartoš and Schimmack [26] for visual intuition.

We extended the z-curve model to incorporate rounded $p$-values and $p$-values reported as inequalities (i.e., $p < 0.001$). We did so in the same way as the Jager and Leek's approach; we implementing interval censored (for rounded $p$-values) and right censored (for $p$-values reported as inequalies) likelihoods versions of the z-curve model,

$$f^{[l,u]}(z; \theta) = \sum_{j=1}^{J} \pi_j f_{j[a,b]}^{[l,u]}(z; \theta_j),$$

where $l$ and $u$ correspond to the lower and upper censoring points on a mixture of truncated folded normal distributions. For example, a $p$-value reported as inequality, $p < 0.001$, has only a lower censoring point at $z$-score corresponding to $p = 0.001$, and rounded $p$-value, $p = 0.02$, has a lower censoring point at a $z$-score corresponding to $p = 0.025$ and upper censoring point at a $z$-score corresponding to $p = 0.015$.

## Simulation study

We performed a simulation study that extends the simulations performed by Jager and Leek [19] in several ways. We did not simulate $\mathcal{H}_1$ $p$-values directly (i.e., we did not use a right skewed beta-distribution as a Jagger and Leek). Instead, we simulated $\mathcal{H}_1$ $p$-values from two-sided $z$-tests. We drew power of each simulated $z$-test (the only required parameter) from a

distribution based on Lamberink and colleagues' [16] empirical estimate of medical clinical trials' power (excluding all power estimates based on meta-analyses with non-significant results). This allowed us to assess the performance of the methods under heterogeneity of power to detect $\mathcal{H}_1$ corresponding to the actual medical literature. To simulate $\mathcal{H}_0$ p-values, we used a uniform distribution.

We manipulated the true false discovery rate from 0 to 1 with a step size of 0.01 and simulated 10,000 observed significant p-values by changing the proportion of p-values simulated from $\mathcal{H}_0$ and $\mathcal{H}_1$. Similarly to Jager and Leek [19], we performed four simulation scenarios with an increasing percentage of imprecisely reported p-values. Scenario A used exact p-values, scenario B rounded p-values to three decimal places (with p-values lower than 0.001 censored at 0.001), scenario C rounds 20% p-values to two decimal places (with p-values rounded to 0 censored at 0.01), and scenario D first rounds 20% p-values to two decimal places and further censors 20% p-values at on of the closest ceilings (0.05, 0.01, or 0.001).

Fig 1 displays the true (x-axis) vs. estimated (y-axis) false discovery rate for Jager and Leek's [19] swfdr method and the false discovery risk for z-curve across the different scenarios (panels). When precise p-values are reported (panel A in the upper left corner), z-curve handles the heterogeneity in power well across the whole range of false discovery rate and produces accurate estimates of false discovery risk. Higher estimate than the actual false discovery rates are expected because the false discovery risk is an estimate of the maximum false discovery rate. Discrepancies are especially expected when power of true hypothesis tests is low. For the simulated scenarios, the discrepancies are less than 20 percentage points and decrease as the true false discovery rate increases. Even though Jager and Leek's [19] method aims to estimate the true false discovery rates, it produces higher estimates than z-curve. This is problematic because the method produces inflated estimates of the true false discovery rate. Even if the estimates were interpreted as maximum estimates, the method is less sensitive to the actual variation in the false discovery rate than the z-curve method.

Panel B shows that the z-curve method produces similar results when p-values are rounded to three decimals. The Jager and Leek's [19] method however experiences estimation issues, especially in the lower spectrum of the true false discovery rate since the current swfdr implementation only allows to deal with rounding to two decimal places (we also tried specifying the p-values as a rounded input; however, the optimizing routine failed with several errors).

Panel C shows a surprisingly similar performance of the two methods when 20% of p-values are rounded to two decimals, except for very high levels of true false discovery rates, where Jager and Leek's [19] method starts to underestimate the false discovery rate. Despite the similar performance, the results have to be interpreted as estimates of the false discovery risk (maximum false discovery rate) because both methods overestimate the true false discovery rate for low false discovery rates.

Panel D shows that both methods have problems when 20% of p-values are at the closest ceiling of .05, .01, or.001 without providing clear information about the exact p-value. Underestimation of true false discovery rates over 40% is not as serious problem because any actual false discovery rate over 40% is unacceptably high. One potential solution to the underestimation in this scenario might by exclusion of the censored p-values from the analyses, assuming that censoring is independent the size of the p-value.

Root mean square error and bias of the false discovery rate estimates for each scenario summarized in Table 1 show that z-curve produces estimates with considerably lower root mean square error. The results for bias show that both methods tend to produce higher estimates than the true false discovery rate. For z-curve this is expected because it aims to estimate the maximum false discovery rate. It would only be a problem if estimates of the false discovery

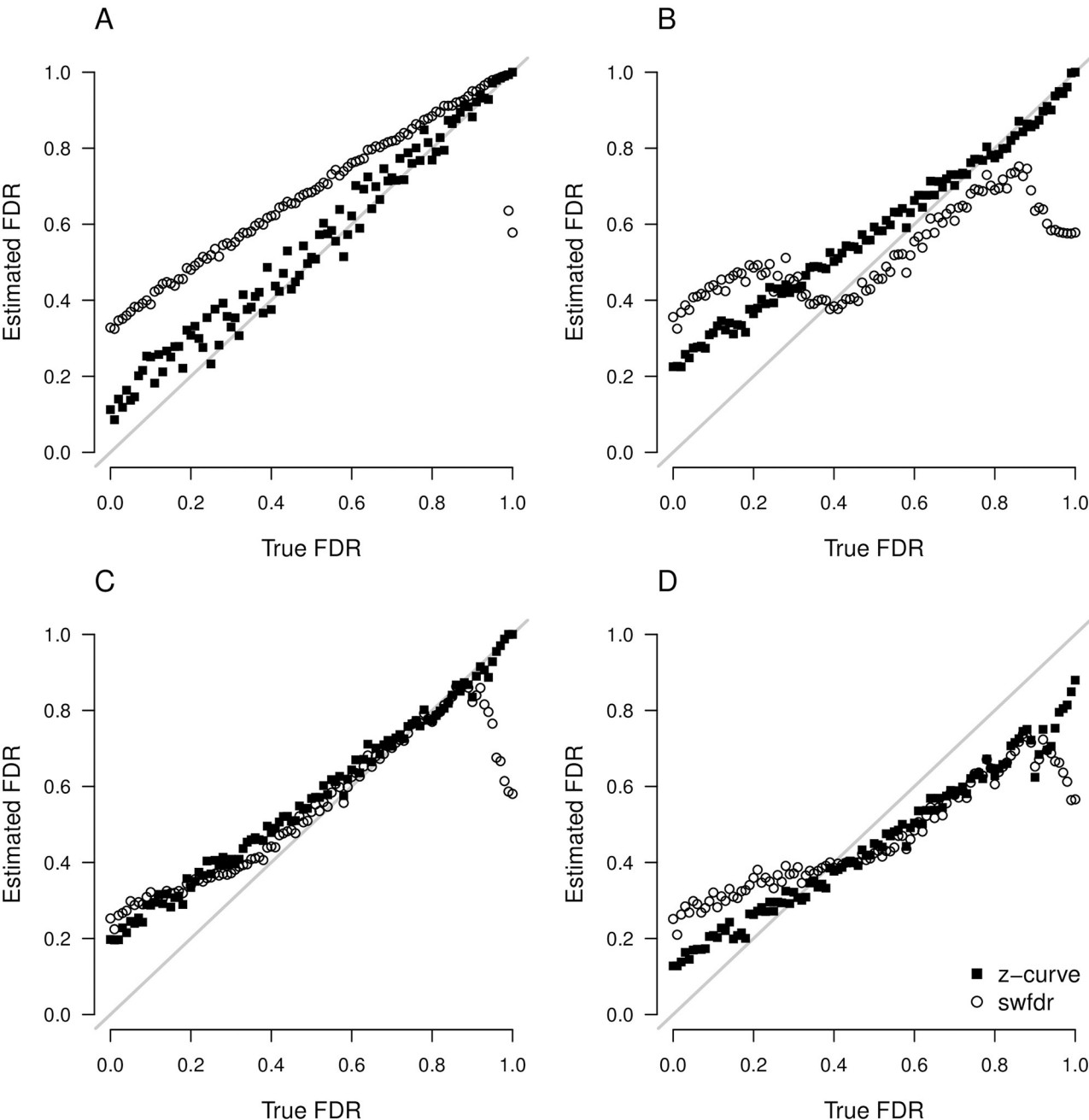

**Fig 1. Comparison of z-curve and Jager and Leek's false discovery risk estimates using a simulation study.** Estimated false discovery rate (FDR, y-axis) for z-curve (black-filled squares) and Jager and Leek's swfdr (empty circles) vs. the true false discovery rate (FDR, x-axis) across four simulation scenarios (panels). Scenario A uses exact *p*-values, scenario B rounds *p*-values to three decimal places, scenario C rounds 20% *p*-values to two decimal places, and scenario D rounds 20% *p*-values to two decimal places and censors 20% *p*-values at on of the closest significance levels (0.05, 0.01, or 0.001).

risk were lower than the actual false discovery rate. This is only the case in Scenario D, but as shown previously, underestimation only occurs when the true false discovery rate is high.

To summarize, our simulation confirms that Jager and Leek's [19] method provides meaningful estimates of the false discovery risk and that the method is more likely to overestimate rather than underestimate the true false discovery rate. Our results also show that z-curve

**Table 1. Root mean square error and bias of z-curve and Jager and Leek's false discovery risk estimates in a simulation study.**

| Method (RMSE) | A | B | C | D |
|---|---|---|---|---|
| z-curve | 0.07 (0.00) | 0.12 (0.01) | 0.10 (0.01) | 0.11 (0.01) |
| swfdr | 0.21 (0.01) | 0.21 (0.01) | 0.13 (0.01) | 0.17 (0.01) |
| Method (bias) | A | B | C | D |
| z-curve | 0.04 (0.00) | 0.09 (0.01) | 0.07 (0.01) | -0.05 (0.01) |
| swfdr | 0.17 (0.01) | 0.02 (0.02) | 0.04 (0.01) | -0.03 (0.02) |

Root mean square error (RMSE) and bias + standard errors (in brackets) for the estimated false discovery rate (FDR) by z-curve and Jager and Leek's swfdr [19] across four simulation scenarios (columns). Scenario A uses exact $p$-values, scenario B rounds $p$-values to three decimal places, scenario C rounds 20% $p$-values to two decimal places, and scenario D rounds 20% $p$-values to two decimal places and censors 20% $p$-values at on of the closest significance levels (0.05, 0.01, or 0.001).

improves over the original method and that the modifications can handle rounding and imprecise reporting when the false discovery rates are below 40%. Only if estimates exceed 40%, it is possible that most published results are false positive results.

## Application to medical journals

We developed an improved abstract scraping algorithm to extract $p$-values from abstracts in major medical journals accessible through https://pubmed.ncbi.nlm.nih.gov/. Our improved algoritm addressed multiple concerns raised in commentaries to the Jager and Leek's [19] original article. Specifically, (1) we extracted $p$-values only from abstracts labeled as "randomized controlled trial" or "clinical trial" as suggested by [20, 21, 23], (2) we improved the regex script for extracting $p$-values to cover more possible notations as suggested by [23], (3) we extracted confidence intervals from abstracts not reporting $p$-values as suggested by [22, 23]. We further scraped $p$-values from abstracts in "PLoS Medicine" to compare the false discovery rate estimates to a less-selective journal as suggested by [20]. Finally, we randomly subset the scraped $p$-values to include only a single $p$-value per abstract in all analyses, thus breaking the correlation between the estimates as suggested by [20]. Although there are additional limitations inherent to the chosen approach, these improvements, along with our improved estimation method, make it possible to test the prediction by several commentators that the false discovery rate is well above 14%.

We executed the scraping protocol on July 2021 and scraped abstracts published since 2000 (see Table 2 for a summary of the scraped data). Interactive visualization of the individual abstracts and scraped values can be accessed at https://tinyurl.com/zcurve-FDR.

Fig 2 visualizes z-curve and Jager and Leek [19] method's false discovery risk estimates based on scraped abstracts from clinical trials and randomized controlled trials and further divided by journal and whether the article was published before (and including) 2010 (left) or after 2010 (right). We see that, in line with the simulation results, Jager and Leek's [19] method produces slightly higher false discovery rate estimates. Furthermore, z-curve produced considerably wider bootstrapped confidence intervals, suggesting that the confidence interval reported by Jager and Leek [19] (± 1 percentage point) was too narrow.

A comparison of the false discovery estimates based on data before (and including) 2010 and after 2010 shows that confidence intervals overlap, suggesting that false discovery rates have not changed. Separate analyses based on clinical trials and randomized controlled trials

**Table 2. Summary of the scraped data.**

| Journal | Abstracts | CT or RCT | Scrapeable | *p*-values |
|---|---|---|---|---|
| Lancet | 7638 | 2099 | 1556 | 5403 |
| BMJ | 4480 | 939 | 643 | 1625 |
| NEJM | 4912 | 2767 | 2304 | 7137 |
| JAMA | 5491 | 1452 | 1274 | 4484 |
| PLoS | 2944 | 346 | 279 | 1102 |
| Total | 25465 | 7603 | 6056 | 19751 |

The "Abstracts" column contains the total number of accessed abstracts, the "CT or RCT" column contains the number of abstracts labels as either "clinical trials" or "randomized controlled trials", the "Scrapeable" column contains the number of further abstracts that contained at least one automatically scrapeable *p*-value or a confidence interval, and the "*p*-values" column contains the total number of *p*-values extracted. PLoS stands for the PLoS Medicine and NEJM stands for the New England journal of medicine.

also showed no significant differences (see Fig 3). Therefore, to reduce the uncertainty about the false discovery rate, we estimate the false discovery rate for each journal irrespective of publication year. The resulting false discovery rate estimates based on z-curve and Jager and Leek's [19] method are summarized in Table 3. We find that all false discovery rate estimates fall within a .05 to .30 interval. Finally, further aggregating data across the journals provides a false discovery rate estimate of 0.13, 95% [0.08, 0.21] based on z-curve and 0.19, 95% [0.17, 0.20] based on Jager and Leek's [19] method. This finding suggests that Jager and Leek's [19] extraction method slightly underestimate the false discovery rate, whereas their model overestimated the false discovery rate. Finally, our improved false discovery risk estimate based on z-curve closely matches the original false discovery rate estimate.

## Additional z-curve results

So far, we used the expected discovery rate only to estimate the false discovery risk, but the expected discovery rate provides valuable information in itself. Ioannidis's [3] predictions of the false discovery rate for clinical trials were based on two scenarios. One scenario assumed high power (80%) and a 1:1 ratio of true and false hypothesis. This scenario implies a true

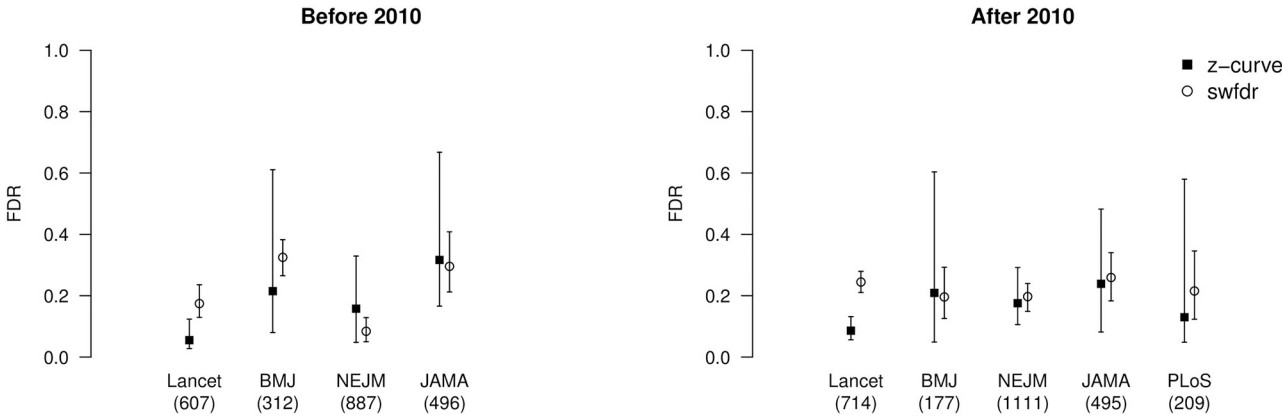

**Fig 2. Estimates of false discovery risk across decades and journals.** Estimated false discovery rate (FDR, y-axis) with z-curve (black-filled squares) and Jager and Leek's [19] swfdr (empty circles) based on clinical trials and randomized control trials divided by journal (x-axis, with information about unique articles in brackets) and whether the article was published before (and including) 2010 (left) or after 2010 (right).

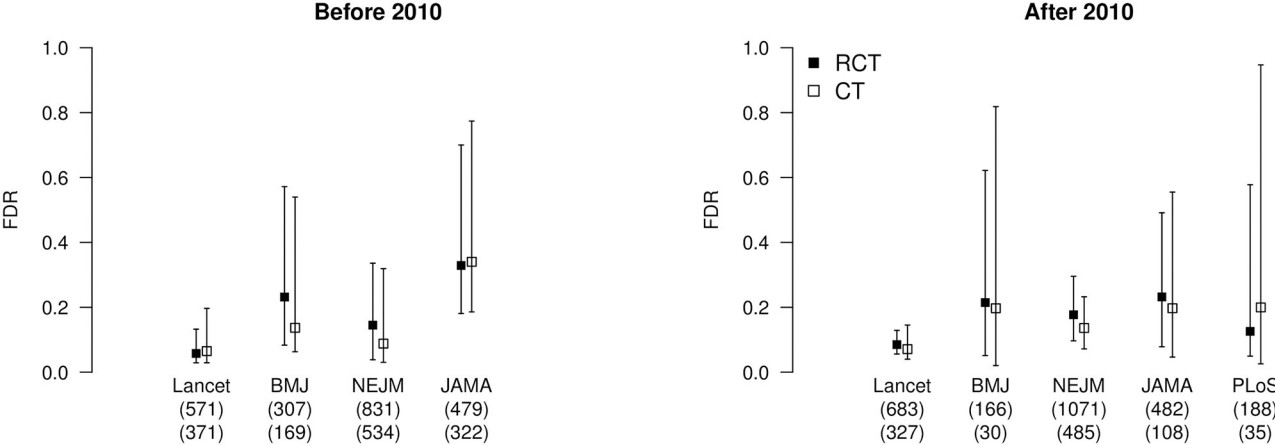

**Fig 3. Estimates of false discovery risk across decades, journals, and study types.** Estimated false discovery rate (FDR, y-axis) for randomized controlled trials (RCT, black-filled squares) and clinical trials (CT, empty squares) with z-curve and divided by journal (x-axis, with information about unique articles in brackets, first line for RCT and second line for CT) and whether the article was published before (and including) 2010 (left) or after 2010 (right).

discovery rate of .5*.8 + .5*.05 = 42.5%. The second scenario assumed 20% power and a ratio of 1 true hypothesis for every 5 false hypotheses. This scenario implies a true discovery rate of .17*.20 + .83*.05 = 7.5%. Scenarios also varied in terms of bias that would inflate the observed discovery rates. However, z-curve corrects for this bias. Thus, it is interesting to compare z-curve's estimates of the discovery rate with Ioannidis's assumptions about the discovery rates in clinical trials. The z-curve estimate of the EDR was 30% with a 95% confidence interval from 20% to 41%. This finding suggests that most published results from clinical trials in our sample were well-powered and likely to test a true hypothesis. This conclusion is also consistent with the finding that our estimate of the false discovery risk is similar to Ioannidis's predictions for high quality clinical trials, FDR = 15%. Thus, when considering top medical journals, a simple explanation for the discrepancy between Ioannidis's claim and our results is that Ioannidis's underestimated the proportion of well-powered studies in the literature.

The expected discovery rate also provides valuable information about the extent of selection bias in medical journals. While the expected discovery rate is only 30%, the observed discovery rate (i.e., the percentage of significant results in abstracts) is more than double (69.7%). This

**Table 3. Combined and journal specific false discovery risk estimates.**

| Journal | z-curve | swfdr |
|---|---|---|
| Lancet | 0.07 [0.05, 0.11] | 0.21 [0.18, 0.24] |
| BMJ | 0.23 [0.08, 0.54] | 0.28 [0.23, 0.32] |
| NEJM | 0.18 [0.10, 0.30] | 0.14 [0.11, 0.18] |
| JAMA | 0.29 [0.15, 0.55] | 0.26 [0.21, 0.31] |
| PLoS | 0.12 [0.05, 0.54] | 0.21 [0.13, 0.32] |
| Combined | 0.13 [0.08, 0.21] | 0.19 [0.17, 0.20] |

False discovery risk (FDR) estimates and 95% confidence interval for each of the analyzed journal and combined data set based on clinical trials and randomized controlled trials published since 2000 with z-curve and Jager and Leek's [19] swfdr method.

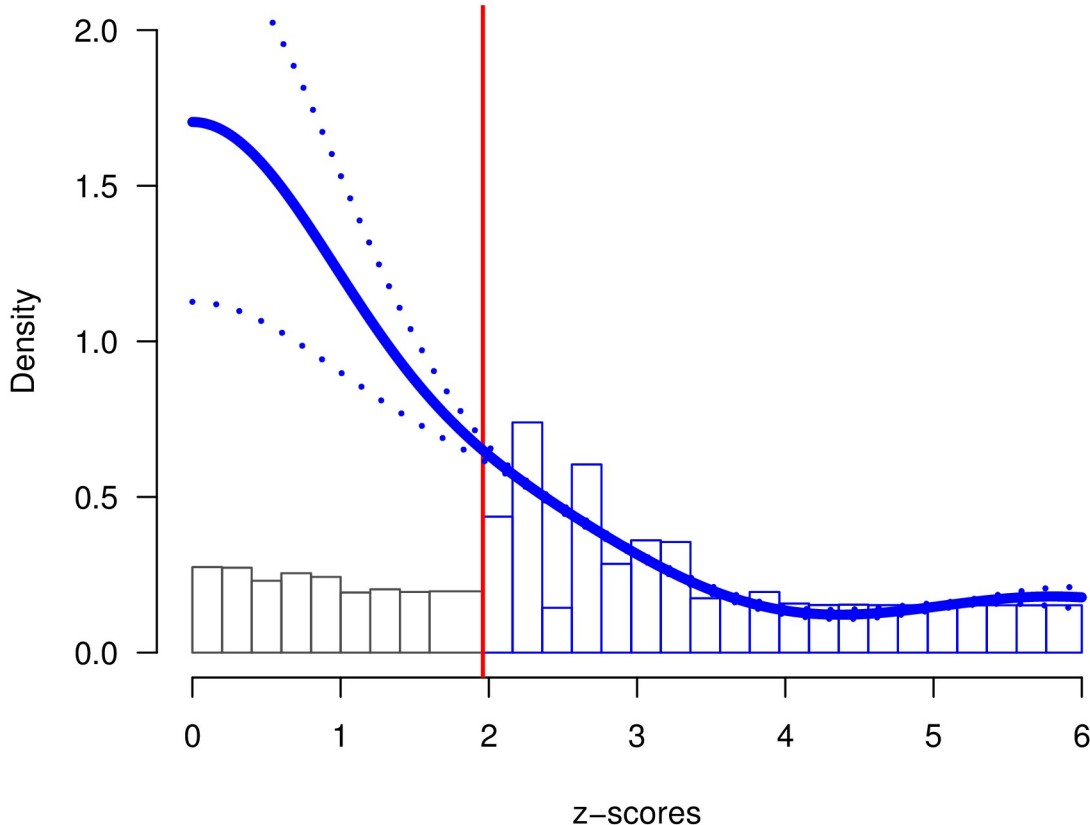

**Fig 4. Z-curve of the combined data set.** Distribution of all significant *p*-values converted into *z*-statistics with fitted z-curve density (solid blue line) and 95% point-wise confidence bands (doted blue line).

discrepancy is visible in Fig 4. The histogram of observed non-significant z-scores does not match the predicted distribution (blue curve). This evidence of considerable selection bias implies that reported effect sizes are inflated by selection bias. Thus, follow-up studies need to adjust effect sizes when planning the sample sizes via power analyses. Moreover, meta-analyses need to correct for selection bias to obtain unbiased estimates of the average effect size [27].

Z-curve also provides information about the expected replication rate of statistically significant results in medical abstracts, defined as the probability of obtaining a statistically significant result in a close replication study with the same sample size and significance criterion (see e.g., [28–30] for alternatives). The expected replication rate is not a simple complement of the false discovery risk, as a true discovery might fail to replicate due to low power of the replication study and the false discovery might "falsely" replicate with a probability corresponding to $\alpha$. The expected replication rate would be 65% with a confidence interval ranging from 61% to 69%. It is important to realize that this expected replication rate includes replications of true and false hypotheses. Although the actual false positive rate is not known, we can use the z-curve estimate to assume that 14% of the significant results are false positives. Under this assumption the power of replication studies of true hypotheses is 75%. In contrast, the probability of obtaining a false positive result again is only 0.7%. Thus, actual replication studies even with the same sample sizes successfully reduce the false positive risk. However, the risk of false negatives is substantial and 44% of non-significant results in replication studies test a true hypothesis.

Furthermore, z-curve estimates of the replication rate are higher than actual replication rates [26]. One reason could be that exact replication studies are impossible and changes in population will result in lower power due to selection bias and regression to the mean. In the worst case, the actual replication rate might be as low as the expected discovery rate. Thus, our results predict that the success rate of actual replication studies in medicine will be somewhere between 30% and 65%. To ensure a high probability of replicating a true result, it is therefore necessary to increase sample sizes of replication studies. When this is not possible, it is important to interpret an unsuccessful replication outcomes with caution and to conduct further research. A single failed replication study is inconclusive.

Many researchers falsely assume that the $\alpha$ = .05 ensures a false discovery risk of only 5%. However, the proportion of false positives equals $\alpha$ only when all tests are performed on true null hypotheses and all results are published. In the real world, the false positive rate depends on the proportion of true null hypotheses and the statistical power of studies examining true alternative hypotheses. We showed that in top medical journals $\alpha$ = .05 entails an FDR of 14%. To lower the FDR to 5% or less, it is necessary to lower $\alpha$. Z-curve analysis can help to find an $\alpha$ level that meets this criterion (by finding $\alpha$ for the desired FDR using Eq 1). With $\alpha$ = .01, the false discovery risk decreases to 4% (consequently, the expected discovery rate would also decreases to 20%). Based on these results, it is possible to use $\alpha$ = .01 as a criterion to reject the null-hypothesis while maintaining a false positive risk below 5%. While journals may be reluctant to impose a lower level of $\alpha$, readers can use this criterion to maintain a reasonably low risk of accepting a false hypothesis.

## Discussion

Like many other human activities, science relies on trust. Ioannidis's (2005) article suggested that trust in science is unwarranted and that readers are more likely to encounter false than true claims in scientific journals. In response to these concerns, a new field of meta-science emerged to examine the credibility of published results. Our article builds on these efforts by introducing an improved method to estimate the false discovery risk in top medical journals, with a focus on clinical trials. Contrary to Ioannidis's suggestion that most of these results may be false positives, we found that the false discovery risk is 14% with the traditional criterion for statistical significance ($\alpha$ = .05) and 4% with a lower threshold of $\alpha$ = .01. Although our results provide some reassurance that medical research published in top journals is more robust than Ioannidis predicted, the false discovery risk in other medical journals and life sciences might be higher than our estimate. Importantly, our results do not warrant blind complacency when conducting and evaluating research or shifting focus from conducting well-designed and highly powered studies.

At the same time, our results also show some problems that could be easily addressed by journal editors. The biggest problem remains publication bias in favor of statistically significant results. The selection for significance has many undesirable consequences. Although medicine has responded to this problem by demanding preregistration of clinical trials, our results suggest that selection for significance remains a pervasive problem in medical research. A novel contribution of z-curve is the ability to quantify the amount of selection bias. Journal editors can use this tool to track selection bias and implement policies to reduce it. This can be achieved in several ways. First editors may publish more well-designed replication studies with non-significant results. Second, editors may check more carefully whether researchers followed their preregistration. Finally, editors can prioritize studies with high sample sizes.

Despite our improvements over Jager and Leel's study, our study has a number of limitations that can be addressed in future research. One concern is that results in abstracts may not

be representative of other results in the results section. Another concern is that our selection of journals is not representative. This are valid concerns that limit our conclusions to the particular journals that we examined. It would be wrong to generalize to other scientific disciplines or other medical journals. We showed that z-curve is a useful tool to estimate the false positive risk and to adjust $\alpha$ to limit the false positive risk to a desirable level. This tool can be used to estimate the false discovery risk for other disciplines and research areas and set $\alpha$ levels that are consistent with the discovery rates of these fields. We believe that empirical estimates of discovery rates, false positive risks, selection bias, and replication rates can inform scientific policies to ensure that published results provide a solid foundation for scientific progress.

## Author Contributions

**Conceptualization:** Ulrich Schimmack, František Bartoš.

**Formal analysis:** František Bartoš.

**Investigation:** Ulrich Schimmack.

**Methodology:** František Bartoš.

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
