## [Decision Letter · Decision Letter 0]

12 Jan 2023

PONE-D-22-04565

Estimating the false discovery risk in medical journals based on published p-values

PLOS ONE

Dear Dr. Schimmack,

Thank you for submitting your manuscript to PLOS ONE. After careful consideration, we feel that it has merit but does not fully meet PLOS ONE’s publication criteria as it currently stands. Therefore, we invite you to submit a revised version of the manuscript that addresses the points raised during the review process.

We look forward to receiving your revised manuscript.

Kind regards,

Niklas Bobrovitz

Academic Editor

PLOS ONE

Journal Requirements:

Additional Editor Comments:

Thank you for submitting your article to PLOS ONE. The reviewers have identified major revisions that need to be considered. I agree with their assessments and encourage you to carefully address each suggestion. 

Reviewers' comments:

Reviewer's Responses to Questions

**Comments to the Author**

1. Is the manuscript technically sound, and do the data support the conclusions?

Reviewer #1: Yes

Reviewer #2: Partly

Reviewer #3: Partly

2. Has the statistical analysis been performed appropriately and rigorously? 

Reviewer #1: I Don't Know

Reviewer #2: N/A

Reviewer #3: Yes

3. Have the authors made all data underlying the findings in their manuscript fully available?

Reviewer #1: Yes

Reviewer #2: Yes

Reviewer #3: Yes

4. Is the manuscript presented in an intelligible fashion and written in standard English?

Reviewer #1: Yes

Reviewer #2: No

Reviewer #3: Yes

5. Review Comments to the Author

Reviewer #1: Minor points:

it is a personal choice, but to me the starting reference to vaccines seems completely unnecessary and rather out of place. If the authors want to make the starting point that science works, they could refer to excellent academic articles like https://doi.org/10.1073/pnas.1711786114.

lines 40-42 it is "well known that" with no reference sounds very unconvincing to me. I would refer to the gigantic systematic reviews by Song et al (and perhaps there is something more recent but I doubt it will be that extensive). However, note that, according to Song et al, the evidence of the kind of problem the authors give for granted is not as firmly established as the authors state. In particular the author confound the issue of the file drawer problem (null results don't get published) with a causal assumption about what generates it (journals reject them), which is not well supported. I would change the sentence to something like "journals may suffer from a publication bias" or "from a selective reporting problem" and cite one or few reviews.

Fairly important concerns:

I think that a deeper hidden assumption in this whole paragraph (and the whole analysis, really) is this dichotomy between "true" and "false" results, which is rather unrealistic. Perhaps I am missing something, but what does it mean to have "100% power"? Power is relative to a magnitude of effect size, and there is always a threshold that needs to be set to determine it. Effects smaller than whatever threshold is set are not "false" and yet will be less detectable. I would encourage the authors to discuss this assumption more, and refer to a recent analysis that - luckily for them - suggests that even relaxing this assumption the picture doesn't change substantially in terms of false discovery rates etc. see https://osf.io/preprints/metaarxiv/jk7sa/

Lines 185 and following: first, the authors need to explain better the link between z-curve and replicability, and particualrly explain what they define as replicability (which is presumably linked to the chances of a replication falling within the confidence interval of the original, or other arbitrary criterion which is based on the same logic of P-values, and is not necessarily the correct criterion (see for example the writings of Gelman on how it wrong to think of replication in binary terms). But more importantly, the authors fail to mention the many replicability studies that, contrary to what they refer, do NOT show unambigiuously low replicability. This was discussed somehow in their reference n. 2 and also in Chapter 4 of this book: https://global.oup.com/academic/product/research-integrity-9780190938550?cc=gb&lang=en&#, but is also discussed in many of those same studies (e.g. experimental economics, experimental philosophy, cognitive science all concluded they do not have much of a problem, and their replicability rates only modestly higher than those in the other studies.

Again, the claim that reproducibility is too low depends on what expectation one should have, which is a matter of defining what standards and thresholds and assumptions one makes about changes in the population and methods etc. The authors mention some of this, but in my opinion they should either develop the argument in full as it deserves, or should remove this part, which once more seems to oversell the implications of the results.

More important concerns:

I feel that the article needs to provide in greater detail, to the general reader, what the methods used imply. The Z curve method deserves its own introduction, and so does the rest of the analysis. I happen to have known the Jager and Leek paper and therefore could grasp at least the generalities of the simulation methods used, but I wonder what the average reader outside Metascience can understand of all this.

There needs to be a formal Methods section that explains the modelling in detail.

The authors need to be much more cautious about their results and interpretation. Apart from the hidden assumptions mentioned above and other modelling choices that may become more apparent once the methods are laid out in full, I can see at least two major critiques of their work, only one of which is discussed, and insufficiently in my opinion. The implications of these critiques are opposite to each other.

On the one hand, abstracts in top journals are likely not representative of the overall population of studies, not just because the methodological standards are likely higher, but also because the types of studies published in these journals are often meeting different assumptions from those made By Ioannidis 2005. I am thinking that many studies will be clinical trials, which are usually carried out when the level of uncertainty is at 50% (i.e. relative high priors). And many other studies in these journals are completing a research program, conclusively proving something that was believed for a long time and had mounting evidence (again, high priors). The priors may be much lower for hypotheses tested in lower-ranking journals. This would entail that their numbers may be grossly under-estimating the "bulk" of the problem.

On the other hand, one could also argue that abstract tend to over-represent small P-values relative to the fulltext they are summarizing, and therefore even these estimates are likely an over-estimate of the true rate of false positives.

Either way, it seems improper for the authors to draw general conclusions about "medical research" or even "top medical journals" the way they do repeatedly in the text.

The discussion opens with a long reflection on science and trust, which I feel slows down the pace of the article and would be better placed at the end of the discussion or in the introduction, if not removed entirely.

I was a bit baffled by the conclusions of the authors who, after having announced in the abstract and suggested in the results that the false discovery rate is much much lower than what many fear, then adopt a dramatic and negative tone at the end, mainly enphasizing the dire implications of their findings. What is the problem with concluding that things may not be so bad after all? Wasn't that the point? And, more importantly, due to limitations that are currently only modestly discussed, the authors should be much more cautious about any conclusion they draw, one way or the other.

In conclusion, I think that this work is certainly interesting and worthy of publication, but in my opinion it needs to be expanded and reshaped, by lowering the tones, avoiding the rhetoric, and instead focusing more on better describing the methods and evaluating the results more critically.

Reviewer #2: The manuscript entitled “Estimating the false discovery risk in medical journals based on published p-values” by Scjimmac and Bartos, presents another attempt at estimating the risk of false discovery in the biomedical sciences. The authors do make some improvements over previous attempts, and I commend them for their rigor. However, I am not sure it really matters if the FDR is 50% or 20% or 12%. What matters is that we understand the factors contributing to a false discovery and train scientists to avoid them. The authors should address the following:

1) The primary contributor to false discoveries in the biomedical sciences is low statistical power. The finding that the risk of false discovery is lower than predicted in a particular sample of a particular subset of the biomedical literature using scraping methods that, by the authors own admission, are prone to error is not really very informative. And it does not change that fact that many studies are underpowered and therefore are likely to indicate false discoveries. My concern is that this manuscript will be viewed as a vindication of current (flawed) research practices. Perhaps rewriting the introduction and discussion to take a more balanced approach to improving the quality of research rather than a somewhat combative approach to disproving previous estimates might be beneficial.

2) On page 7 the authors state that their estimate of the success of actual replication studies will be somewhere between 30 and 65%. Shouldn’t the false discovery risk be commensurate with these values since a failure to replicated would be considered to indicate a false discovery? It is not clear how these 2 parameters (replication success and initial probability of a false discovery) could be disconnected.

3) The suggestion on page 8 that the alpha value could be adjusted to reduce the risk of false positives is not very good advice. A more stringent alpha level applied to underpowered studies will lead to an inflation of effect size. Wouldn’t it be better to simply suggest researchers appropriately power their studies?

4) The manuscript is mostly well written, however there are some sentences that are hard to parse and appear to be missing a word or two. For example:

Page 2 under Simulation Study: “…,we used estimated power individual studies based…”

Page 4, first full paragraph: “…since the current implementation only allows to deal with…”

Page 4, 3rd full paragraph: “…underestimation problem might to exclude p-values…”

Reviewer #3: The authors aim to estimate the false discovery risk (the maximum false discovery rate) in medical literature, by analyzing p-values reported in PubMed abstracts. The authors approach seems sound, and I would like to applaud the fact that the authors have made all their code and study artifacts publicly available. However, I do have some concerns, which I hope the authors can address:

Major:

On page 5, the authors mention that they only extract p-values from abstracts labeled as “randomized clinical trial” or “clinical trial”. I think this is an important qualification that is currently easy to miss. Arguably, clinical trials are less likely to have publication bias and p-hacking, because they are expensive (and therefore less likely to be buried), often must be pre-registered, and typically require oversight by regulators. In fact, if I regenerate Figure 4 using data from my field (observational research), the cutoff at z = 1.96 is far more extreme than what the authors found for trials. I think this nuance is so important it should be mentioned in the title of the manuscript.

I think the organization of the manuscript can be greatly improved. Currently, there is no distinction between introduction and methods, and the authors directly dive into simulations without explaining what their approach is that they’re evaluating in simulations. The authors currently assume the readers are completely knowledgeable of what was written in their z-curve paper, and the paper by Jagger and Leek. It would be good to summarize what was in these papers, and not explain what the authors did just by explaining how their approach differs from Jagger and Leek (since most readers, like me, will have forgotten the details of that paper).

Minor:

Page 2, line 48: the term ‘z-curve’ needs to be introduced. I can guess it is the method referred to in the previous sentence, but that should be made explicit.

Page 2, line 82: “we used estimated power individual studies based on medical meta-analysis.” I’m not sure what the authors mean by this. Also, I think this sentence is grammatically incorrect.

Page 3, line 68: “We manipulated the true false discovery rate from 0 to 1…”. Please explain how.

Page 5, line 142.5: “the ‘Extractable’ column” Note that this column is called ‘Scrapeable’ in Table 2.

6. PLOS authors have the option to publish the peer review history of their article (what does this mean?). If published, this will include your full peer review and any attached files.

Reviewer #1: No

Reviewer #2: No

Reviewer #3: **Yes: **Martijn Schuemie

---

## [Decision Letter · Decision Letter 1]

5 Jun 2023

PONE-D-22-04565R1Estimating the false discovery risk of (randomized) clinical trials in medical journals based on published p-valuesPLOS ONE

Dear Dr. Schimmack,

Thank you for submitting your manuscript to PLOS ONE. After careful consideration, we feel that it has merit but does not fully meet PLOS ONE’s publication criteria as it currently stands. Therefore, we invite you to submit a revised version of the manuscript that addresses the points raised during the review process.

Dr. Schimmack, Thank you for re-submitting your manuscript and for your patience during this review process. One of the reviewers has judged your manuscript to now be acceptable for the publication however, the other reviewer feels that their comments were not adequately addressed. Please could you review their comments and either incorporate the suggested revisions or provide clear justification for not doing so. You may even wish to discuss in your manuscript some of the critiques that the reviewer has flagged - they have raised some significant concerns about the messages that they are deriving from the manuscript.  ==============================

We look forward to receiving your revised manuscript.

Kind regards,

Niklas Bobrovitz

Academic Editor

PLOS ONE

Reviewers' comments:

Reviewer's Responses to Questions

**Comments to the Author**

1. If the authors have adequately addressed your comments raised in a previous round of review and you feel that this manuscript is now acceptable for publication, you may indicate that here to bypass the “Comments to the Author” section, enter your conflict of interest statement in the “Confidential to Editor” section, and submit your "Accept" recommendation.

Reviewer #1: All comments have been addressed

Reviewer #2: (No Response)

2. Is the manuscript technically sound, and do the data support the conclusions?

Reviewer #1: Yes

Reviewer #2: Partly

3. Has the statistical analysis been performed appropriately and rigorously? 

Reviewer #1: Yes

Reviewer #2: Yes

4. Have the authors made all data underlying the findings in their manuscript fully available?

Reviewer #1: Yes

Reviewer #2: Yes

5. Is the manuscript presented in an intelligible fashion and written in standard English?

Reviewer #1: Yes

Reviewer #2: Yes

6. Review Comments to the Author

Reviewer #1: (No Response)

Reviewer #2: 1) I was a bit surprised that the authors took a simple statement regarding the magnitude of the false discovery rate to not matter and turned into an abandonment of statistical significance testing. I must have not been as clear as necessary on making this point, so I will try to carefully reiterate and hopefully provoke a less defensive response.

I believe that we would agree that there is currently a lack of statistical training and rigor in many branches of the life sciences. Underpowered studies, which I and most researchers define as less than 80% power are common, leading to more false discoveries than expected. Let’s set aside the magnitude of this effect for the moment and focus on the authors statement made in their response that “consumers of scientific information want to have some assurance that they can trust published results to be more than just statistical noise.” Indeed, we agree on this. However, as a consumer of scientific information myself, I would feel (and am) much more comfortable reading a journal that requires the authors to describe statistical power and how it was derived than I would be reading a journal that just assumes all is well because the FDR is only 20%, not 60%.

The authors seem to be arguing with Ioannidis in their response. While his work may have been more theoretical, and therefore require certain assumptions, it cannot be denied that his intentionally provocative and seminal article has led to an increased interest in statistical rigor and has, as the authors state, led to the emergence of the field of meta-science in which they are currently working. Given this, I maintain that it is irresponsible to publish an analysis of false discovery risk predicted in a particular sample of a particular subset of the biomedical literature using scraping methods that, by the authors own admission, are prone to error without reinforcing the importance of conducting well powered studies. I strongly advise the authors to consider adding a few lines to their discussion simply making that clear.

2) The authors point on alpha of 0.05 vs 0.01 is confusing and needs to be better stated or removed. Despite their rebuttal statement that they “do not propose that researchers change their a priori alpha level”, they clearly seem to think that they should. They state that “To lower the FDR to 5% or less it is necessary to lower alpha” and imply a change in the a priori alpha value by stating “while journals may be reluctant to impose a lower level of alpha…” The authors fail to recognize the issue of effect size inflation and impact of this on future replication studies. The fact that “many researchers falsely assume that the alpha = .05 ensures a false discovery risk of only 5%” is again a problem in statistical training and rigor. Wouldn’t it be far better to suggest that readers actually understand what the alpha value means rather than suggest they calibrate to some personal alpha level? Choosing to ignore a result from a properly powered study with alpha =0.03 is bad advice and not good statistical practice.

7. PLOS authors have the option to publish the peer review history of their article (what does this mean?). If published, this will include your full peer review and any attached files.

Reviewer #1: No

Reviewer #2: No

---

## [Author Response · Author response to Decision Letter 1]

15 Jun 2023

please, see attached pdf response

---

## [Decision Letter · Decision Letter 2]

2 Aug 2023

Estimating the false discovery risk of (randomized) clinical trials in medical journals based on published p-values

PONE-D-22-04565R2

Dear Dr. Schimmack,

We’re pleased to inform you that your manuscript has been judged scientifically suitable for publication and will be formally accepted for publication once it meets all outstanding technical requirements.

Kind regards,

Niklas Bobrovitz

Academic Editor

PLOS ONE

Additional Editor Comments (optional):

Thank you for your patience during the review process. 

Reviewers' comments:

Reviewer's Responses to Questions

**Comments to the Author**

1. If the authors have adequately addressed your comments raised in a previous round of review and you feel that this manuscript is now acceptable for publication, you may indicate that here to bypass the “Comments to the Author” section, enter your conflict of interest statement in the “Confidential to Editor” section, and submit your "Accept" recommendation.

Reviewer #2: All comments have been addressed

2. Is the manuscript technically sound, and do the data support the conclusions?

Reviewer #2: Yes

3. Has the statistical analysis been performed appropriately and rigorously? 

Reviewer #2: Yes

4. Have the authors made all data underlying the findings in their manuscript fully available?

Reviewer #2: Yes

5. Is the manuscript presented in an intelligible fashion and written in standard English?

Reviewer #2: Yes

6. Review Comments to the Author

Reviewer #2: (No Response)

7. PLOS authors have the option to publish the peer review history of their article (what does this mean?). If published, this will include your full peer review and any attached files.

Reviewer #2: No

---

## [Editor Report · Acceptance letter]

7 Aug 2023

PONE-D-22-04565R2 

Estimating the false discovery risk of (randomized) clinical trials in medical journals based on published *p*-values 

Dear Dr. Schimmack:

I'm pleased to inform you that your manuscript has been deemed suitable for publication in PLOS ONE. Congratulations! Your manuscript is now with our production department. 

Kind regards, 

on behalf of

Dr. Niklas Bobrovitz 

Academic Editor

PLOS ONE